# From dyadic coping to emotional sharing and multimodal interpersonal synchrony: Protocol for a laboratory experiment

Zihao Zeng[1,2,3]*, Karen Holtmaat[2,3,4], Xihan Jia[2,3], Annet Kleiboer[2,3], Francesca Rhighetti[5], Anne-Marie Brouwer[6], Fabian Ramseyer[7], Sophie C.F. Hendrikse[8], Sander L. Koole[2,3]

1 School of Educational Science, Hunan Normal University, Changsha, China, 2 Department of Clinical, Neuro and Developmental Psychology, Vrije Universiteit Amsterdam, Amsterdam, The Netherlands, 3 Amsterdam Public Health, Mental Health, Amsterdam, The Netherlands, 4 Cancer Center Amsterdam, Treatment and Quality of Life, Amsterdam, The Netherlands, 5 Department of Experimental and Applied Psychology, Vrije Universiteit Amsterdam, Amsterdam, The Netherlands, 6 Netherlands Organisation for Applied Scientific Research TNO, Soesterberg, The Netherlands, 7 Department of Clinical Psychology and Psychotherapy, University of Bern, Bern, Switzerland, 8 Center of Research on Psychological and Somatic Disorders, Tilburg University, Tilburg, The Netherlands

* zhzeng@hunnu.edu.cn, z.zihao@vu.nl

## Abstract

During interpersonal emotion regulation, relationship partners mutually regulate each other's emotional states. Interpersonal emotion regulation occurs at three main timescales: phasic (from several hundred milliseconds to about 10s), tonic (from 10s to 1 hour), and chronic (from weeks to months and years). Prior research has examined interpersonal emotion regulation at only one or two timescales simultaneously. The proposed research will examine variables relating to interpersonal emotion regulation in close relationships across all three timescales. A total of 150 romantic couples will engage in an emotional sharing task, in which they will be instructed to either engage in natural sharing or co-rumination. At the phasic timescale, primary outcomes will be interpersonal synchrony in movements and cardiovascular responses throughout the sharing task. At the tonic timescale, primary outcomes will be changes in mood and emotional appraisals pre- and post-sharing. At the chronic timescale, the study will primarily assess individual differences in relationship quality and dyadic coping style prior to the task, which are expected to shape phasic and tonic patterns during emotional sharing. Our general expectation is that phasic patterns in interpersonal emotion regulation (e.g., movement synchrony) will be meaningfully related to tonic patterns (e.g., mood change), which, in turn, will be meaningfully related to chronic patterns (e.g., relationship quality). More differentiated hypotheses and exploratory analyses are detailed in the protocol. The results of this research will contribute to the integration of interpersonal emotion regulation theories across different time scales.

**Data availability statement:** No datasets were generated or analysed during the current study. All relevant data from this study will be made available upon study completion.

**Funding:** This article was facilitated by scholarship of the Chinese Scholarship Council (202206720004) and Youth Program of The Hunan Provincial Natural Science Foundation (2025JJ60165) to Zihao Zeng and NWO Open Competition (406.18.GO.024) to Sander L. Koole. The funders had no role in study design, data collection and analysis, decision to publish, or preparation of the manuscript.

**Competing interests:** The authors have declared that no competing interests exist.

## Introduction

Within close relationships, partners regulate each other's emotional states. Such interpersonal emotion regulation is of vital significance for the health and wellbeing of the individual relationship partners, as well as for the quality of their relationship [1–3]. It thus seems important to learn more about the processes that underlie interpersonal emotion regulation within close relationships.

One fundamental aspect of interpersonal emotion regulation is that it unfolds dynamically over time [1,4]. Prior work has distinguished between three main timescales of interpersonal emotion regulation [5–7]. First, the phasic timescale runs from several hundreds of milliseconds to about 10 seconds. Interpersonal emotion regulation at the phasic timescale is mostly nonverbal, and occurs in exchanges of facial expressions, eye gaze, breathing patterns, or whole-body movements [7]. Second, the tonic timescale runs from about 10 seconds to roughly one hour. Interpersonal emotion regulation at the tonic timescale consists of socially meaningful, often verbal exchanges, such as emotional sharing [8], for example during a casual encounter or a psychotherapy session [9]. Third and last, the chronic timescale runs from several weeks to months and years. Interpersonal emotion regulation at the chronic timescale consists of more or less stable patterns of behaviour, which may be driven by personal dispositions (e.g., chronic attachment style [10]) and/or stable relationship characteristics (e.g., dyadic coping styles [11]). Importantly, in this study, the chronic timescale is not assessed longitudinally after the task, but is operationalized through individual and relational characteristics (e.g., dyadic coping and relationship quality) measured before the emotional sharing interaction, which are hypothesized to moderate the effects observed at the phasic and tonic levels.

At any given moment, the functioning of a close relationship is characterized by phasic, tonic, and chronic processes, along with the interplay of these processes. Leading theories on interpersonal emotion regulation primarily focus on one of these time scales. Synchrony [5] focuses on spontaneous moment-to-moment interpersonal alignment of aspects like bodily movements and physiological processes. Theories on emotional sharing [8] take the time frame of a conversation as their focus, and theories on dyadic coping [12] often investigate partners' coping behaviours over a more extended period of time. To date, research has examined variables relating to interpersonal emotion regulation at only one, or at most two timeframes. In the proposed experiment, we seek to obtain a more comprehensive picture of interpersonal emotion regulation, by examining variables relating to interpersonal emotion regulation in close relationships across all three of the phasic, tonic, and chronic timeframes.

### Proposed experiment and hypotheses

In the proposed experiment, a sample of couples (i.e., romantic partners) will engage in an emotional sharing task [13]. In this task, we ask each of the relationship partners twice to share a negative emotional experience and provide each other with emotional support. By means of verbal instructions, we will manipulate whether the couple engages in natural sharing or co-rumination (2*2 design). Co-rumination is characterized by a prolonged focus on problems and negative emotions. It has a twofold impact:

talking about problems and emotions fosters companionship, social support and positive relationship quality, to a greater extent than natural sharing [5]. After a single episode of co-rumination, sharers may feel better, due to the possibility to extensively share their feelings and to their experience of bonding with the supporter. While co-rumination is often associated with emotional intimacy, long-term engagement has been linked to greater psychological distress [14]. This distinction underlines the relevance of our current experimental approach, which integrates multiple outcome domains across timescales. This is likely due to the ruminative nature of co-rumination, in which little cognitive changes, solutions or new insights occur [15,16].

While earlier research [17] emphasized the affiliative functions of nonverbal synchrony, recent studies suggest that such effects may not be robust or consistent across contexts. For example, DiGiovanni et al. (2024) found no significant increase in physiological synchrony in co-rumination compared to natural sharing [18], and Lin et al. (2023) observed no clear association between behavioral synchrony and perceived support [19]. These findings call for a more nuanced approach that considers boundary conditions and relational moderators. Against this backdrop, our study examines the phasic and tonic emotional and physiological effects of co-rumination compared to natural emotional sharing. Rather than assuming uniform benefits or detriments, we hypothesize that co-rumination may shape interpersonal processes in complex ways: potentially facilitating interpersonal synchrony and positive mood, while simultaneously yielding fewer cognitive shifts in emotional appraisal. We further explore how these effects may be moderated by relationship quality and dyadic coping style.

Compared to the effects of the co-rumination manipulation on synchrony, mood and appraisal change, we expect that couples' relationship quality and their dyadic coping strategies have a much stronger effect on synchrony, mood and appraisal change. Couples with a high relationship quality and constructive dyadic coping strategies will already have employed a range of interpersonal emotion regulation and coping strategies to deal with the problem they are discussing, and therefore these brief episodes of natural sharing or co-rumination will minimally affect their synchrony, mood and appraisal changes. In contrast, for couples with a poorer relationship quality or dyadic coping strategies, the co-rumination manipulation to engage in extensively discussing the sharers problems and negative emotions, will have a greater effect on their synchrony and positive mood change than when they engage in their natural way of emotional sharing, but will still result in minimal appraisal changes.

Prior to the emotional sharing task, we will measure a set of variables related to the chronic timescale: relationship quality, dyadic coping and a range of secondary variables (e.g., attachment style, empathy and perceived social support). During the emotional sharing task itself, we will assess phasic processes in the form of movement synchrony [5] and cardiovascular synchrony [20]. Before and after the emotional sharing tasks, we will measure tonic processes: mood and appraisal changes and a range of secondary variables (e.g., conversation experience, perception of partners' responsiveness and feelings of closeness with the partner).

The proposed experiment has both confirmatory and exploratory parts. The confirmatory part is based on hypotheses that were explicitly derived from the literature on interpersonal emotion regulation. Because of the complexity of the study, we have grouped our hypotheses according to their respective timescales (see Table 1). More detailed information can be found in *Supplementary 1 in* S1 File.

Moreover, we will conduct a series of exploratory analyses involving the secondary variables to further inform our understanding of interpersonal emotion regulation within close relationships. These exploratory analyses will probe the intricate connections between relational factors, individual characteristics, emotional sharing and physiological responses. In addition, in a second set of exploratory analysis we will be looking at synchrony in verbal communication: 1) linguistic synchrony; 2) vocal synchrony; 3) facial synchrony. These exploratory analyses are elaborated in Supplementary *1 in* S1 File.

## Methods and statistical analysis

### Design

This study has a mixed factorial-experimental nested design. In total, each couple engages four times in the emotional sharing task, twice in the co-rumination and twice in the natural sharing condition. Each participant will twice share a

**Table 1. Main confirmatory hypotheses for the three timescales.**

| Timescale | Hypotheses |
|---|---|
| *Phasic* | 1. Relationship partners will display significant levels of movement and cardiovascular synchrony during the emotional sharing tasks, relative to a relevant baseline level of randomized pseudo-interactions.<br>2. Compared to natural sharing, co-rumination may be associated with enhanced movement and cardiovascular synchrony. |
| *Tonic* | 3. Emotional sharing will generally make sharers and supporters feel better, leading them to report positive mood changes.<br>4. Emotional sharing will generally help sharers to reappraise their feelings, leading them to report changes in emotional appraisals.<br>5. Compared to natural sharing, co-rumination may lead to equal or even more positive mood outcomes, possibly due to increased emotional closeness and perceived support.<br>6. Compared to natural sharing, co-rumination will be associated with smaller changes in emotional appraisals. |
| *Chronic* | 7. Higher relationship quality and more constructive dyadic coping strategies will be associated with 1) stronger movement synchrony; 2) stronger cardiovascular synchrony; 3) stronger positive mood changes; 4) more emotional appraisal changes.<br>8. Higher relationship quality and constructive dyadic coping may buffer or attenuate potential differences between co-rumination and natural sharing in: (a) movement synchrony, (b) cardiovascular synchrony, (c) mood responses, and (d) emotional appraisal changes. |

distressing experience and will twice act as supporter. Both aspects will be experimentally manipulated between participants. Measures of mood and appraisal will be nested within participants, while movement and cardiovascular synchrony will be nested within dyads. The chronic variables (i.e., dyadic coping and relationship variables) will be used correlational.

## Participants

Couples aged 18–65 in a romantic relationship of at least 4 months will be eligible for participation. Participants also must be proficient in the Dutch language. Exclusion criteria are the use of psychoactive medications and a history of neurological conditions. The data collection for the ongoing study is actively underway, with participant recruitment expected to be completed by 31/12/2024. Recruitment for this study began on 01/01/2024 and will end on 31/12/2024.

## Sample size determination

We determined the sample size based on a priori power analysis (G*Power) [21]. For a power of at least 80% to compare the means of two independent groups, a minimum sample size of 128 dyads is required to detect medium effects (0.5) at $\alpha < .05$ for two-tailed comparisons. A minimum sample size of 59–114 dyads is required to detect small effects (0.1–0.2) at $\alpha < .05$ for linear multiple regression. We aim to recruit a sample size of 150 dyads for our study, so that we will be able to accommodate potential exclusions of participants due to unforeseen reasons. Compared to other studies on synchrony [22,23], our study features a large sample size.

## Procedure

Participants will be recruited partly through a recruitment agency and partly by research assistants spreading flyers on the university campus, making announcements to the participant pool of the psychology program at the Vrije Universiteit Amsterdam and by recruiting from their social networks. Interested individuals will be asked to complete a brief online questionnaire to assess their eligibility. To reduce random error in psychophysiological responses an email will be sent to the participants two days before the experimental session, with the request to refrain from alcohol consumption and intense physical activity 24 hours before the session, and caffeine consumption 2 hours before the session.

Participants will arrive as a couple and will be personally greeted by a research assistant at the building entrance and escorted to the laboratory. A diagram of the laboratory is shown in Fig 1 and a photograph of the lab space is shown in Fig 2. Upon arrival at the lab, the couple will be greeted by a second research assistant. Each participant will be led to their individual workspace. Next, the experimental procedures will be explained to the participants by the research assistants

*Diagram of the Interaction Laboratory and Observation Room*

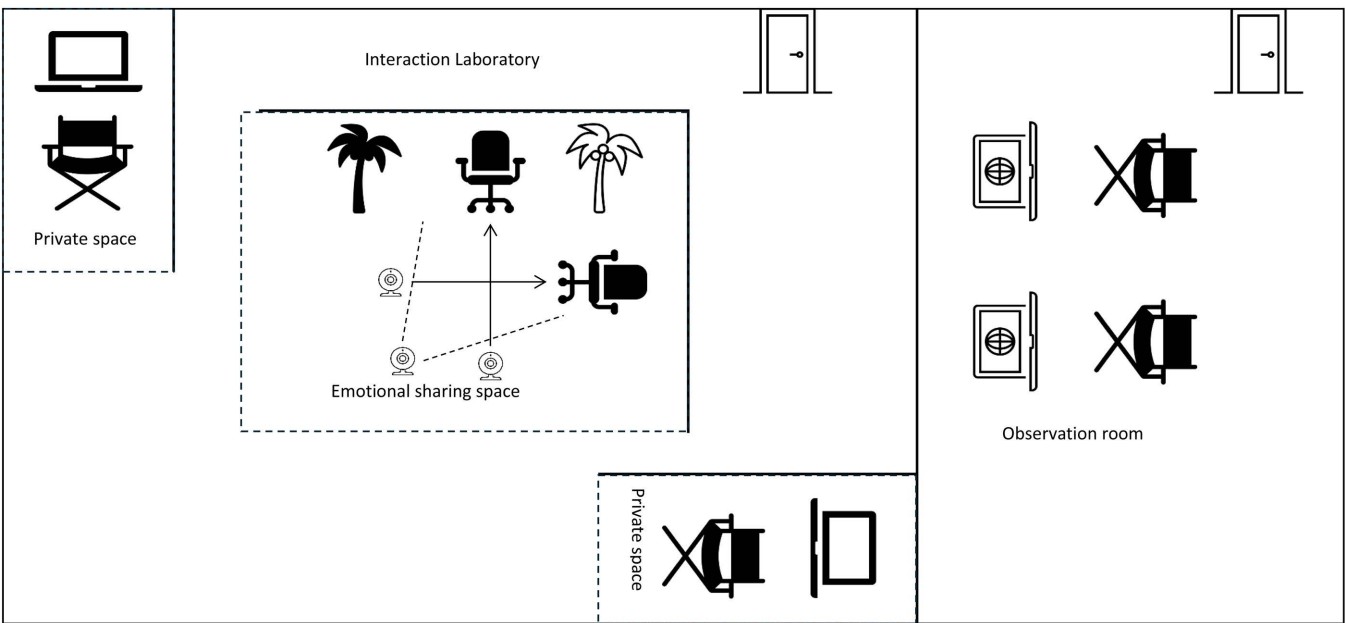

**Fig 1. Diagram of the interaction laboratory and observation room.**

and written informed consent will be obtained. After these explanations, participants will verify their compliance with the behavioural instructions. Divergences of the behavioural instructions will be noted and included as covariates to control for their effects. To do so, we will record participants' alcohol consumption (yes or no), their physical activity intensity (5-point score, from 'none' to 'very strong') and caffeine digestion (1 mug to 5 mugs).

In the first phase of the experiment, participants will be asked to rate their mood on the mood adjective check-list (BEF) [24]. Then they will be instructed to generate two personal negative experiences using the Experience-Generation Questionnaire (EGQ) [25]. Subsequently, the Experience Scales (ES) [26] will be used to tap into their appraisal of these two experiences. Next, participants will be fitted with the sensors of the Vrije Universiteit - Ambulatory Monitoring System (VU-AMS). Once the VU-AMS has been set up, participants will be requested to complete the pre-sharing survey.

Subsequently, participants will be provided with instructions for the emotional sharing task, consisting of four dialogues. Upon completion of the entire emotional sharing task, participants will complete a post-sharing survey and the ECG device will be removed. Finally, participants will be debriefed, rewarded, and thanked. Fig 3 shows the whole experimental session. In total, the experiment takes approximately 2 hours, with task instructions taking 10 minutes, device fitting taking 15 minutes, the pre-sharing survey taking 15 minutes, the emotional sharing task for four episodes lasting 60 minutes, the post-sharing survey taking 15 minutes, and the device removal and closure taking 5 minutes.

## Experiences generation task

In the emotional sharing tasks, participants will be asked to talk about two negative experiences from their own life. To illustrate what is meant by negative experiences, the ERQ lists seven negative experiences, such as having an argument with a close friend or family member. Participants will be asked to write down two negative experiences they are currently facing. By focusing on personal experiences, we ensure that the study has high personal relevance for the participants.

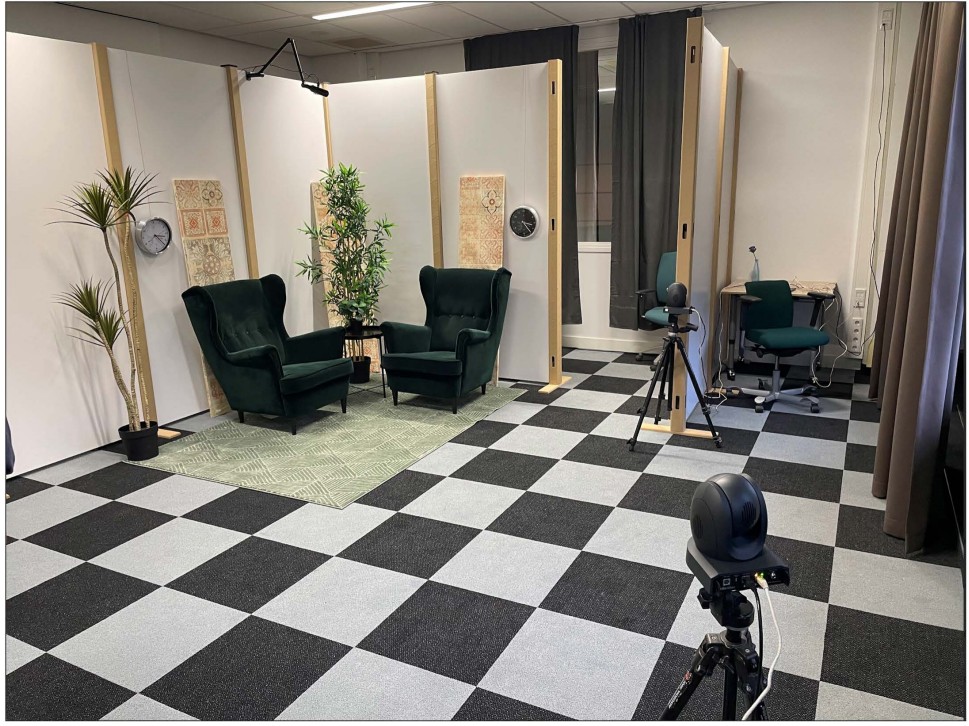

**Fig 2. Photograph of the emotional sharing space and one individual workspace.**

### Emotional sharing task

The emotional sharing task, which is modelled after Nils & Rimé (2009), involves a structured procedure to make participants share their emotions in a controlled experimental setting [13]. Participants will be having a total of four eight-minutes episodes with their partner to share the two experiences they each wrote down in the experience generation task. During each episode, one participant will act as a sharer (to discuss the problem) and the other participant will act as a supporter (to listen and respond to the sharer). The roles of sharer and supporter will be randomly alternated so that each participant will be a sharer and a supporter twice.

### Co-rumination manipulation

We will experimentally manipulate the instructions of the emotional sharing task, following a validated procedure developed by Tudder et al (2023) [27]. For half of the sharing episodes, the supporter will receive instructions that promote co-rumination, which involves focusing on a topic, repeatedly discussing the issues, speculating about the antecedents and consequences of the issues, and exploring and unearthing negative emotions. For the other sharing episodes, the supporter will be instructed to respond in a natural way to the sharer. The order of the co-rumination versus natural response will be counterbalanced. Each member of the couple will be instructed once to engage in co-rumination and once to engage naturally with the sharer in all four conditions, the participants will receive digital instructions and physical cards to remind them of their role and the associated instructions.

Before the first sharing episode, participants will be taken by their experimenter to the emotional sharing space with the two armchairs (see Fig 1b), so that they can familiarize themselves with this space Then they will be taken back to their

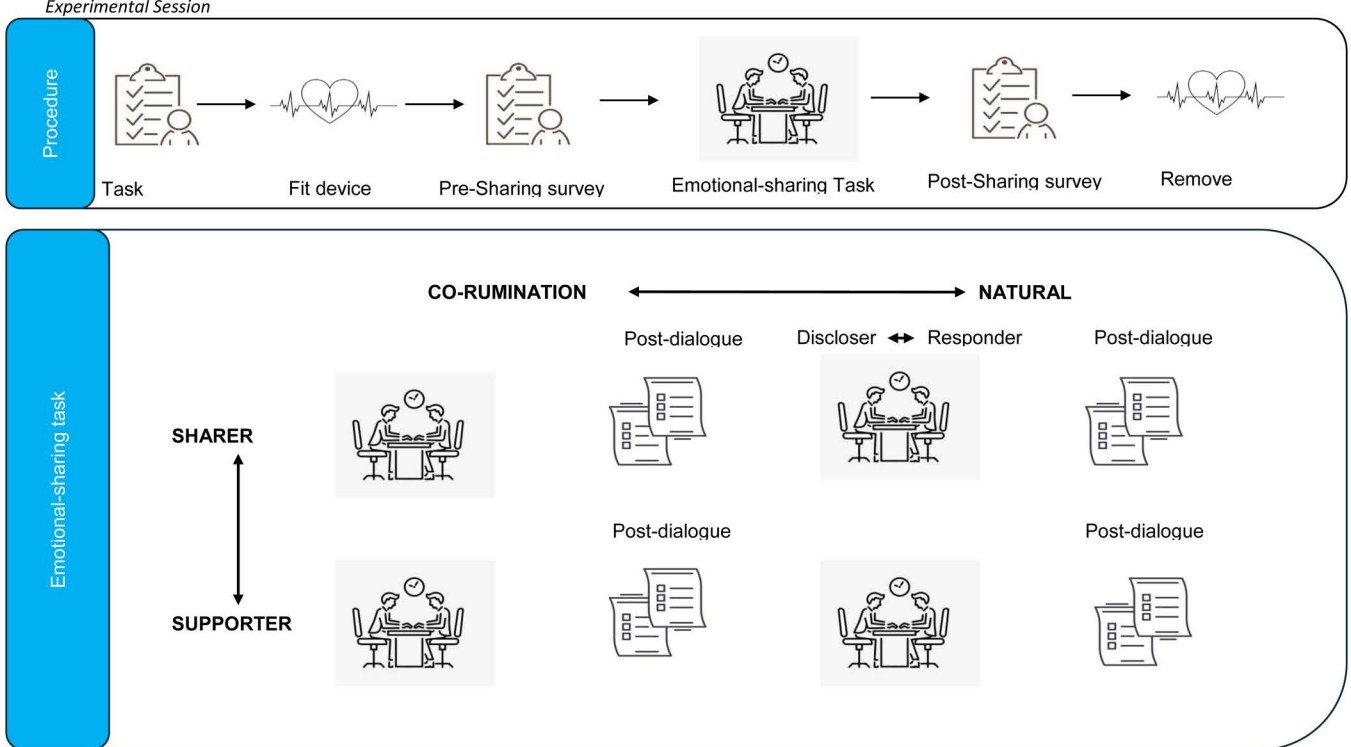

**Fig 3. Experimental session.**

private space, where they will be informed about the first experience to address and be asked to "collect their thoughts" for two minutes. Following this, each participant will again be taken to the emotional sharing space. Participants will then engage in their first round of emotional sharing on the selected experience. After completing the first emotional sharing episode, the participants will go back to the private space and complete the post-dialogue survey about their sharing experience. After completing the first instruction, the instructions for natural sharing and co-rumination will be reversed, and after two rounds, the roles of the sharer and supporter will be switched, and vice versa.

## Ethical considerations

The protocol obtained approval from the Scientific and Ethical Review Board, Vrije Universiteit Amsterdam (protocol number VCWE-2023-130R1). All participants will provide written informed consent. During the data collection process, all study-related documents (data and informed consent forms) will be securely stored in Yoda (Your Data is a cloud storage at SURF and suitable for storing large-scale and sensitive datasets). After completion of the study, final storage and archiving of the electronic data will take place in a dedicated archive at Vrije Universiteit Amsterdam and will be processed in compliance with the General Data Protection Regulation (GDPR). In accordance with legal regulations from the European Union, the data will be stored for a period of 10 years.

## Measurement

Table 2 provides a comprehensive overview of the tools and objectives for the measurement employed in this study. More detailed information can be found in Supplementary 2 in S1 File.

## Data analysis plan

**Phasic timescale.** ***Video recordings and movement synchrony:*** Video recordings of all sharing episodes will be made using three fixed cameras at a frame rate of 25 frames per second. One camera will capture the entire scene of emotional sharing, while the other two will be placed in front of the participants, and their feeds will be connected through split-screen. Before initiating the Motion Energy Analysis (MEA), the split-screen video will allow an independent analysis of each individual to derive objective motion quantification.

The MEA program [28] employs a frame-differencing algorithm, which requires both a static camera-position as well as a stable background setting. Under these conditions, the difference in grayscale pixels between consecutive video frames reflects the bodily movements of participants, which is then quantified into time series of movement. The areas for this kind of quantification can be defined in the program, and we will use a singular region of interest (ROI) encompassing the entire head and upper body of each participant. Based on one ROI per participant, the MEA program will process the video images and generate one CSV file per video containing the time series of raw pixel variations within the designated ROIs. Adjustments for minimal thresholds of movement will be processed in the MEA program according to the criteria suggested by the creator of the program [28]. Following the generation of raw movement data, the rMEA package in R Studio will be employed for further analysis [29].

We will follow the standard pre-processing steps as outlined by Ramseyer and Tschacher (2011) [9] and Tschacher et al. (2018) [30]. Initially, a 0.5-second moving average filter will be applied to smooth the time series, reducing fluctuations caused by signal distortions present in the video data. To account for varying sizes of the ROI, the data will be z-transformed, and extreme outliers will be eliminated. Here, we will use the default threshold provided by the authors of rMEA, thereby excluding extreme values higher than 10 times the standard deviation. Data thus filtered and corrected are then subjected to windowed cross lagged cross-correlation analyses for the quantification of nonverbal synchrony.

Specifically, in each 8-minute dialogue, the motion energies of both participants will be cross-correlated in non-overlapping window segments of 30 seconds. The choice of a 30-second window size (750 frames) is to account for the shorter turn-taking latencies in question-centered discussions, consistent with previous work [30]. Windowed cross lagged cross-correlation of up to 5 seconds of positive and negative time lags will be computed with a step size of 0.1 seconds, achieved by progressively shifting one time series relative to the other [9]. The resulting matrix of cross-correlations will then be transformed using Fisher's r-to-Z transformation, and their absolute values will be aggregated over the entire 8-minute interval of the dialogue, resulting in a shared global value of nonverbal synchrony for every sharing episode [9].

To ascertain whether the synchrony detected via MEA exceeded what might occur by chance (i.e., pseudosynchrony), we will engage a specific RMEA function designed for generating pseudointeractions through automated surrogate algorithms. This function shuffles the days in the data and calculates all possible motion energy interactions between couples in the dataset. Subsequently, it dissociates the original pairings, followed by the extraction of a predefined amount of unique MEA datasets, each selected just once to avoid duplication.

To test Hypothesis 1, we will examine whether relationship partners show greater synchrony than pseudo-dyads during the emotional sharing tasks. Following recommendations by Helm et al. (2018) [31] and DiGiovanni et al. (2024) [18], we will compute windowed cross-correlations to generate time series of movement synchrony. We will then apply multilevel modeling (MLM) using the *lme4* package in R to compare synchrony scores between real dyads and matched pseudo-dyads, accounting for the nested structure of time windows within dyads. To test Hypothesis 2, we will examine whether co-rumination leads to greater synchrony than natural sharing. Movement synchrony data will be modeled using multilevel models with time windows (Level 1) nested within participants (Level 2) and dyads (Level 3). Experimental condition (co-rumination vs. natural sharing) will be entered as a fixed effect predictor, and dyad ID and participant ID will be modeled as random effects. This approach accounts for both the repeated nature of the synchrony time series and the interdependence between partners in a dyad.

**Table 2. Overview of the measurement tools, objectives and measurement period.**

| Timescale | Name | Measurement objectives | Measurement period |
|---|---|---|---|
| *Phasic* | VRMS | Movement synchrony | Emotional sharing episodes |
| | CMS | Cardiovascular synchrony | Emotional sharing episodes |
| *Tonic* | BEF | Mood (e.g., *"Helplessness."*) | Pre-sharing survey and post-dialogue surveys |
| | ES | Emotional Appraisal (e.g., *"How intense were the negative emotions that you had during this experience?"*) | Pre-sharing survey and post-dialogue surveys |
| | IOSS | How close the respondent feels with another person (seven images) | Pre-sharing survey and post-dialogue surveys |
| | SOE | The experiences of partners in conversation (e.g., *"I felt completely like myself during the conversation."*) | Post-dialogue surveys |
| | PPR | Individuals' perceptions of their partners' responsiveness (e.g., *"My partner understood me."*) | Post-dialogue surveys |
| | IER-DSFP | Different interpersonal emotion regulation strategies from partner (e.g., *"My partner tried to get me to talk over and over about what is bothering me."*) | Post-dialogue surveys |
| | RQ-ASSC | Relationship quality (e.g., *"My partner and I agreed on how I could best approach the problem."*) | Post-dialogue surveys |
| *Chronic* | DCI | Dyadic coping strategies (e.g., *"I tell my partner that it is not that bad and help him/her to see the situation in a different light."*) | Pre-sharing survey |
| | PQ-SF | Quality of romantic relationships (e.g., *"He/she takes me in his arms."*) | Pre-sharing survey |
| | PSAS-SF | Attachment styles (e.g., *"It helps to turn to my partner in times of need."*) | Pre-sharing survey |
| | TAS-20 | Alexithymia (e.g., *"I am often confused about what emotion I am feeling."*) | Pre-sharing survey |
| | OSRS | Participants' perception of emotion regulation (e.g., *"I gave someone helpful advice to try to make them feel better."*) | Pre-sharing survey |
| | SSL | Perceived social support received (e.g., *"Did you ask for help or advice?"*) | Pre-sharing survey |
| | SV | Sociodemographic variables (e.g., *"How old are you."*) | Post-sharing survey |
| | UCLA-LS | Loneliness (e.g., *"I lack companionship."*) | Post-sharing survey |
| | IRI | Empathy (e.g., *"I sometimes find it difficult to see things from the other person's perspective."*) | Post-sharing survey |
| | DDI | Distress disclosure (e.g., "*When I feel upset, I usually confide in my friends*") | Post-sharing survey |
| | BSRI | Sex role (e.g., " *Assess the extent to which the following characteristics apply to you: loving*") | Post-sharing survey |
| | MHC | Total mental health (one item " *How happy are you?*")) | Post-sharing survey |

VRMS*:* Video recordings and movement synchrony, CMS: Cardiovascular measurement and synchrony, BEF: Mood adjective checklist, ES: Experience scales, IOSS: Inclusion of other in the self scale, SOE: Shearer/supporter' own experience, PPR: Perceived partner responsiveness, IER-DSFP: Interpersonal emotion regulation with different strategies from partner, RQ-ASSC: Relationship quality: agreement, support, self-disclosure, closeness, TAS-20: The Toronto alexithymia-20 scale, PQ-SF: Partnership questionnaire—short form, PSAS-SF: Partner-specific attachment security short form, DCI: Dyadic coping inventory, OSRS: The others and self-emotion regulation scale, SSL: Social support list, UCLA: University Of California, Los Angeles, UCIA-LS: UCLA Loneliness Scale, IRI: Interpersonal reactivity index, DDI: Distress disclosure index, BSRI: Bem sex role inventory, MHC: Mental health continuum

***Cardiovascular measurement and synchrony:*** To collect electrocardiography data, the VU-AMS 7 from Vrije Universiteit Amsterdam will be used. This instrument consists of five non-invasive electrodes placed on the participant's chest and back. First, by employing automated and visual data cleansing techniques, potentially anomalous R-wave peaks will undergo manual rectification or will be marked as artifacts to ensure the integrity of the inter-beat interval (IBI) series. IBI is defined as the time (in ms) between two consecutive R-peaks in the ECG signal and is automatically scored by the VU-AMS 7 software. Subsequently, the IBI time series will be resampled at a frequency of 10 Hz, and samples will be divided into epochs of 2000 ms with fixed onset and offset to achieve accurate temporal alignment of IBI values within

dyads. The selection of a 2000 ms epoch length is based on the minimum time required for reliable estimation of heart rate [32]. For each epoch, the average IBI will be calculated, resulting in a time series of epoch-averaged values for each participant.

For each 2000 ms epoch time series, a second-order polynomial regression will be calculated to remove linear and quadratic trends from the data [33]. To eliminate the autocorrelation properties of the signal, the partial autocorrelation function (PACF) of the detrended IBI time series, (i.e., the residuals after polynomial fitting) will be plotted and analysed. First, the autoregressive integrated moving average (ARIMA) model will be applied to the residuals of the polynomial regression, incorporating one autoregressive term, one moving average term, integrated noise, and the residuals obtained from the analysis as inputs into the cross-correlation calculation.

Finally, the resulting IBI time series will be then further processed in Matlab (The MathWorks, Inc., Natick, MA). Specifically, we will use the Matlab functions 'parcorr' for the PACFs, 'armax' for ARIMA modelling and 'xcorr' (with zero-time lag) for the calculation of the cross-correlations. It is important to note that the two measurement methods used to compute IBI synchrony do not measure the distance at which individual heartbeats occur within the dyad simultaneously. Instead, they assess how the heart rate variability between consecutive 2000 milliseconds epochs co-fluctuate across the entire dyad.

To assess whether cardiovascular synchrony exceeds what would be expected by chance (i.e., pseudosynchrony), we will follow the approach of Feldman et al. (2011) [34] and more recent adaptations by DiGiovanni et al. (2024) [18]. We will first calculate windowed cross-correlations on the interbeat interval (IBI) time series for each dyad during the interaction, using time-lagged windows to account for dynamic shifts in coupling. To test whether the observed synchrony exceeds random coupling, we will generate a surrogate dataset of randomly paired pseudo-dyads, and then use multilevel modeling (MLM) to compare synchrony scores between real and pseudo-dyads. This model will account for the nested structure of time windows within dyads and the repeated nature of physiological data. To test Hypothesis 2, we will use MLM to compare cardiovascular synchrony between the co-rumination and natural sharing conditions, including condition as a fixed effect and random intercepts for dyad and participant where appropriate. This approach allows us to properly model interdependence between partners and repeated time-based observations within each interaction episode.

**Tonic timescale.** *Mood change analysis:* To assess changes in mood over time (Hypothesis 3), we will use linear mixed-effects models (LMMs), treating time (pre vs. post) and condition (co-rumination vs. natural sharing) as fixed effects, and including random intercepts for participants nested within dyads to account for the non-independence of dyadic partners and repeated measurements. This approach enables us to examine overall mood changes without splitting sharers and supporters into separate analyses.

For Hypothesis 5, we will examine whether mood following co-rumination differs from mood following natural sharing using the same LMM framework, focusing on post-sharing mood scores across conditions.

*Emotional appraisal analysis:* To evaluate changes in emotional appraisals (Hypothesis 4), we will calculate change scores by subtracting the pre-sharing appraisal rating from the post-sharing appraisal rating for each emotional episode. These change scores will then be entered into linear mixed-effects models, with condition as a fixed effect and random intercepts for participants nested within dyads. This approach accounts for within-person and within-dyad dependence and avoids the use of separate t-tests for each condition. Hypothesis 6 will be tested by comparing appraisal change scores between the co-rumination and natural sharing conditions using the same dyadic LMM framework.

**Chronic timescale.** *Relationship quality and dyadic coping analysis:* We will employ regression analysis to examine the associations between relationship quality and dyadic coping strategies and movement and cardiovascular synchrony (*Hypothesis 7.1 and 7.2*). Additionally, to analyse whether dyadic coping and relationship quality are associated with mood changes we will add these variables as fixed factors in the linear mixed models built for testing hypothesis 3 on mood change over time (*Hypothesis 7.3*). Linear regression analyses will also be used to assess whether relationship quality and dyadic coping are related to emotional appraisal change (*Hypothesis 7.4*).

To assess whether co-rumination versus natural sharing influences the relationship between higher relationship quality and more constructive dyadic coping and movement and cardiovascular synchrony moderated regression analyses will be used (Hypotheses 8.1 and 8.2). To analyse whether the associations between relationship quality and dyadic coping with mood change are associated with effects of co-rumination versus natural sharing we will extend the linear mixed models used for testing hypothesis 7.3. The full model will consist of a random intercept for participants, fixed factors of assessment time and condition, their interaction, and mood as the dependent variable (*Hypothesis 8.3*). Moderated linear regression analyses will be used to assess whether better relationship quality and dyadic coping are associated with weaker effects of the co-rumination condition versus natural sharing on emotional appraisal change (*Hypothesis 8.4*).

The hypotheses presented in Table 1 and the data analysis techniques utilized constitute the core elements of our research. Additionally, we have identified and measured a range of secondary variables that could potentially influence these associations. Our aim is to incorporate these variables as covariates in the analyses to enhance our understanding of the findings. For all statistical tests $\alpha \le 0.05$ (two-tailed) will be applied. Given the wide range of statistical tests performed, there is a concern regarding multiple comparisons. Therefore, to reduce the likelihood of Type I errors, we will apply the Bonferroni correction to all previously mentioned test results.

*Exploratory analyses summary:* In our study, we plan to conduct exploratory analyses to investigate interpersonal synchrony in verbal communication:

1. **Verbal expressions and word use:** We will analyze the frequency and context of specific words used during emotional sharing to see how they correlate with mood and personality changes, and relationship quality.

2. **Vocal arousal coding:** Vocal characteristics such as arousal and pitch variations will be examined to understand their correlations with emotional states and personality traits during interactions.

3. **Facial expression coding:** We will employ facial expression coding to study the emotional responses visible during the sharing tasks, assessing their relationship with mood change and personality traits.

## Discussion

The planned research seeks to comprehensively evaluate interpersonal emotion regulation across phasic, tonic, and chronic time scales concurrently. This research design will allow for the examination of three leading theories of interpersonal emotion regulation—namely, synchrony [5], emotional sharing [8], and dyadic coping [12]. These theoretical frameworks and associated processes have so far not been studied within a single research design. By so doing, the planned research will allow us to examine how moment-to-moment (phasic) processes in interpersonal emotion regulation are related to specific episodes (tonic) of interpersonal emotion regulation, and how these, in turn, are related to more enduring (chronic) patterns in interpersonal emotion regulation.

The planned study -like all research- has limitations. First, the study's generalizability may be limited due to its inclusion of only Dutch-speaking participants with a limited age range and relationship duration. Replicating the planned study with more diverse samples will be important to test the external validity of the findings. Second, the present study contains an experimental manipulation of one tonic process (i.e., co-rumination), but not of phasic processes (movement synchrony and cardiovascular synchrony) or chronic processes, which are both merely measured. Thus, the present study cannot draw causal conclusions about the role of phasic and chronic processes in interpersonal emotion regulation. Third, the study is limited by its reliance on retrospective self-reports for chronic processes and the potential participant fatigue from completing four emotional sharing conversations. While our within-subject design enhanced comparability across conditions, future studies could consider between-subjects or longitudinal designs to reduce participant burden and allow for more direct testing of chronic interpersonal processes over time.

Despite these caveats, the proposed study will be the first to study processes of interpersonal emotion regulation jointly across phasic, tonic, and chronic time scales. We hope that this study will facilitate the development of an integrated theoretical framework for understanding interpersonal emotion regulation.

## Supporting information

**S1 File. Supplementary 1: Proposed Experiment and Hypotheses Supplementary 2: Measurement** .
(DOCX)

## Author contributions

**Conceptualization:** Zihao Zeng, Karen Holtmaat, Annet Kleiboer, Sander L. Koole.

**Methodology:** Zihao Zeng, Karen Holtmaat, Xihan Jia, Annet Kleiboer, Fabian Ramseyer, Sophie C.F. Hendrikse, Sander L. Koole.

**Writing – original draft:** Zihao Zeng.

**Writing – review & editing:** Zihao Zeng, Karen Holtmaat, Francesca Rhighetti, Anne-Marie Brouwer, Fabian Ramseyer, Sophie C.F. Hendrikse, Sander L. Koole.

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
