## [Decision Letter · Decision Letter 0]

26 Mar 2025

PONE-D-24-48074

From Dyadic Coping to Emotional Sharing and Multimodal Interpersonal Synchrony: Protocol for a Laboratory Experiment

PLOS ONE

Dear Dr. Zeng,

Thank you for submitting your manuscript to PLOS ONE. After careful consideration, we feel that it has merit but does not fully meet PLOS ONE’s publication criteria as it currently stands. Therefore, we invite you to submit a revised version of the manuscript that addresses the points raised during the review process.

We look forward to receiving your revised manuscript.

Kind regards,

Rachel Low

Academic Editor

PLOS ONE

Journal Requirements:

“This article was facilitated by a scholarship of the Chinese Scholarship Council (202206720004) to Zihao Zeng and NWO Open Competition Grant 406.18.GO.024 to Sander L. Koole.”

Additional Editor Comments:

Because the reviewer’s comments are clearly expressed, I will not reiterate all of the points that they have raised. To facilitate your revision, here is what I view as the most important issues that you need to address in the revision.

1.    My primary concern is the use of t-tests to analyze the data for behavioural synchrony, as noted by the reviewer. As these data are nested within dyads, it would be more appropriate to use multilevel models so that you account for the dependence within dyadic data.

2.    It appears that there are some relevant and recent co-rumination research that are missing from the manuscript (e.g., Lin et al., 2023). Please make sure to integrate more recent research into your study protocol.

3.    Relatedly, as pointed out by the reviewer, it was surprising that the authors hypothesized that co-rumination would be associated with better mood. Although previous work suggests that co-rumination could lead to better relational outcomes, co-rumination is generally associated poorer emotional outcomes. Please clarify.

Reviewers' comments:

Reviewer's Responses to Questions

**Comments to the Author**

1. Does the manuscript provide a valid rationale for the proposed study, with clearly identified and justified research questions?

Reviewer #1: No

2. Is the protocol technically sound and planned in a manner that will lead to a meaningful outcome and allow testing the stated hypotheses?

Reviewer #1: Partly

3. Is the methodology feasible and described in sufficient detail to allow the work to be replicable?

Reviewer #1: Yes

4. Have the authors described where all data underlying the findings will be made available when the study is complete?

Reviewer #1: Yes

5. Is the manuscript presented in an intelligible fashion and written in standard English?

Reviewer #1: Yes

6. Review Comments to the Author

You may also provide optional suggestions and comments to authors that they might find helpful in planning their study.

Reviewer #1: I was very excited to review this proposal/paper, as this is a topic I am super interested in and think that this multiple timescale approach is an interesting and important one. I commend the authors for conducting such an intensive study with many moving parts! However, I think a number of significant issues with the review of the literature, support for the hypotheses, and the planned analyses. Since data collection is already completed, this is more of an issue. My detailed comments are below.

- The proper citation for the co-rumination manipulation is Tudder et al., 2023. Lin et al., used these data, but the manipulation was validated in Tudder et al., 2023

- There needs to be a more thorough review on the co-rumination literature. There are a lot more, newer papers that examine co-rumination in a similar context to what you are looking at (e.g., co-rumination in adult and romantic relationships).

- For example, although currently just a preprint, a paper by DiGiovanni et al uses state space grids to show that co-rumination is actually related to feeling like problems are MORE solved. This is counter to one of your main hypotheses. Although Rose and colleagues hypothesize co-rum doesn't help solve problems, this is not frequently tested, to my knowledge.

- For your hypothesis regarding relationship quality and synchrony, you should look to Lin et al., 2023 and DiGiovanni et al., 2024 that both explored these topics. Lin looked at behavioral synchrony and digiovanni looked at physiological synchrony. DiGiovanni and colleagues did not find support for more synchrony in the co-rumination condition compared to the natural condition. This is in contrast to your hypotheses and should be integrated into your intro and discussion and the support for your hypotheses.

- It was not clear until a while into the introduction that the chronic timescale would not be assessed as a follow up to the conversation, but instead as variables that affect that in-person conversation. This should be made clearer.

- Why do you hypothesize that there will be better mood after co-rumination? Tudder and colleagues did not find this in their paper using the same exact method. The literature on this generally points to worse mood, however digiovanni et al 2021 have shown this to be heterogeneous and the pre-print I mention above that uses state space grids showed no associations with negative mood. So, although there is some support for what you are saying, most co-rum work shows that co-rum is related to worse mood. This needs to be fleshed out more why you would expect this effect.

- Why not look at any type of relationship outcomes in the tonic phase, like closeness or responsiveness? After reading, it looks like you have those variables, but not hypotheses about them. Why? Or maybe you plan to look at this, but didn't include that in the manuscript.

- I worry about having participants engage in 4 conversations. Two conversations is already taxing enough, and I dont think it is a good idea to have people do both co-rum and natural sharing. I would have suggested to keep people in one condition. However, it seems that data collection is finished now. I might instead analyze data only for the first condition for all participants.

- Why run t-tests for behavioral synchrony? Since there should be moment by moment synchrony, you need to run multilevel models to account for the dependence in the data; that is, there are multiple timepoints and measurements for each dyad. Moreover, you need to account for the dependence between people (sharers and supporters) within a dyad. You shouldn't split up those individuals and run separate analyses. I have the same comment for the IBI series. T-tests are not appropriate to run, unless I am missing something meaningful here about what you are testing (which is possible!). My understanding, though, is that you need a more complicated model that can assess the synchrony across the study, and take into account the repeated nature of the data and the dyadic non-independence. If you want to do non-directional synchrony, you should follow Helm et al., 2018 and digiovanni et al., 2024 (synchrony paper). Or, you might want to run more of an APIMs model, and could look at Thorson et al., 2018

- Same comments as above for the tonic timescale: I don't believe that t-tests are the most appropriate statistical test. You also should be using models that don’t require you to separate out responders and disclosers, but instead using dyadic models that allow you to combine these assessments and account for the dyadic nature of the data.

7. PLOS authors have the option to publish the peer review history of their article (what does this mean? ). If published, this will include your full peer review and any attached files.

**Do you want your identity to be public for this peer review?** For information about this choice, including consent withdrawal, please see our Privacy Policy .

Reviewer #1: No

---

## [Author Response · Author response to Decision Letter 1]

2 Apr 2025

Dear Editor and Reviewers,

Thank you for providing us with the opportunity to revise our manuscript for the PLOS One. We are grateful for the valuable and constructive feedback provided during the first round of reviews, which has significantly improved the clarity and quality of our manuscript.

In this second round of revisions, we have carefully addressed each of the comments. Below, we provide a detailed point-by-point response, outlining the changes made in the manuscript. We greatly appreciate the efforts of the editor and reviewers and yourself in guiding us through this process, and we hope the revised version meets the journal's expectations. Thank you for considering our work, and we look forward to your further feedback.

Sincerely,

Zihao

Reviewer #1:

I was very excited to review this proposal/paper, as this is a topic I am super interested in and think that this multiple timescale approach is an interesting and important one. I commend the authors for conducting such an intensive study with many moving parts! However, I think a number of significant issues with the review of the literature, support for the hypotheses, and the planned analyses. Since data collection is already completed, this is more of an issue. My detailed comments are below.

Our response: Thank you very much for your positive feedback and your thoughtful critique. We appreciate your recognition of the study’s importance and complexity. We provide detailed, point-by-point responses to your specific comments below.

- The proper citation for the co-rumination manipulation is Tudder et al., 2023. Lin et al., used these data, but the manipulation was validated in Tudder et al., 2023

Our response: Thank you for pointing this out. We have corrected the citation accordingly: the co-rumination manipulation is now properly attributed to Tudder et al., 2023, as the original source of validation.

- There needs to be a more thorough review on the co-rumination literature. There are a lot more, newer papers that examine co-rumination in a similar context to what you are looking at (e.g., co-rumination in adult and romantic relationships).

Our response: We thank the reviewer for highlighting the need to provide a more comprehensive and up-to-date review of the co-rumination literature. In response to this helpful suggestion, we have carefully reviewed recent studies that investigate co-rumination in adult and romantic relationships, including work by DiGiovanni et al. (2021, 2024), Lin et al. (2023), and Tudder et al. (2023). These studies explore both the interpersonal and physiological consequences of co-rumination and offer important insights into the variability of its effects depending on context and relationship type.

- For example, although currently just a preprint, a paper by DiGiovanni et al uses state space grids to show that co-rumination is actually related to feeling like problems are MORE solved. This is counter to one of your main hypotheses. Although Rose and colleagues hypothesize co-rum doesn't help solve problems, this is not frequently tested, to my knowledge.

Our response: We thank the reviewer for this insightful comment and for highlighting the recent preprint by DiGiovanni et al. We agree that this work provides a novel and important contribution by reconceptualizing co-rumination as a dyadic and dynamic system using state space grids.

In response, we have revised the introduction sections to explicitly acknowledge this emerging perspective. In particular, we now discuss the DiGiovanni et al. (preprint) study in relation to our own conceptual framework, noting that while co-rumination has typically been associated with interpersonal closeness and intrapersonal distress, its effect on perceived problem solving may depend on specific interactional dynamics (e.g., reciprocal engagement, role symmetry, temporal structure). Revised statement:

“While co-rumination is often associated with emotional intimacy, long-term engagement has been linked to greater psychological distress (Rose, 2021). Recent findings (DiGiovanni et al., preprint) suggest that co-rumination may also enhance perceived problem resolution, raising questions about the conditions under which it may serve adaptive versus maladaptive functions.”

We also clarify that our hypothesis regarding co-rumination. We appreciate the reviewer’s attention to this nuanced issue and believe the manuscript is now strengthened by integrating this newer work and its implications.

DiGiovanni, A. M., Peters, B. J., Li, X., Tudder, A., & Gresham, A. M. (2024). It takes two to co-ruminate: Examining co-rumination as a dyadic and dynamic system [Preprint]. PsyArXiv. https://doi.org/10.31234/osf.io/sgvx3

Rose, A. J. (2021). The costs and benefits of co‐rumination. Child Development Perspectives, 15(3), 176-181.

- For your hypothesis regarding relationship quality and synchrony, you should look to Lin et al., 2023 and DiGiovanni et al., 2024 that both explored these topics. Lin looked at behavioral synchrony and digiovanni looked at physiological synchrony. DiGiovanni and colleagues did not find support for more synchrony in the co-rumination condition compared to the natural condition. This is in contrast to your hypotheses and should be integrated into your intro and discussion and the support for your hypotheses.

Our response: We appreciate the reviewer’s insightful suggestion to consider recent studies by Lin et al. (2023) and DiGiovanni et al. (2024), which directly relate to our hypotheses regarding relationship quality and synchrony. We have carefully reviewed these studies and integrated their findings into both our Introduction sections to better contextualize our work. Specifically, Lin et al. (2023) found no significant association between behavioral synchrony and perceived friendship support, while DiGiovanni et al. (2024) reported that physiological synchrony was not greater in the co-rumination condition compared to natural emotional sharing. These findings indeed challenge our original assumption that co-rumination would consistently amplify synchrony.

In the revised Introduction, we now more clearly acknowledge that prior findings on co-rumination and synchrony are mixed. While earlier research (e.g., Vacharkulksemsuk & Fredrickson, 2012) emphasized the affiliative functions of nonverbal synchrony, the emerging literature suggests that such effects may not be robust or uniform. This perspective strengthens the rationale for examining moderating factors and deepens the conceptual scope of our work.

Revised statement:

“While earlier research (e.g., Vacharkulksemsuk & Fredrickson, 2012) emphasized the affiliative functions of nonverbal synchrony, recent studies suggest that such effects may not be robust or consistent across contexts. For example, DiGiovanni et al. (2024) found no significant increase in physiological synchrony in co-rumination compared to natural sharing, and Lin et al. (2023) observed no clear association between behavioral synchrony and perceived support. These findings call for a more nuanced approach that considers boundary conditions and relational moderators. Against this backdrop, our study examines the phasic and tonic emotional and physiological effects of co-rumination compared to natural emotional sharing. Rather than assuming uniform benefits or detriments, we hypothesize that co-rumination may shape interpersonal processes in complex ways: potentially facilitating interpersonal synchrony and positive mood, while simultaneously yielding fewer cognitive shifts in emotional appraisal. We further explore how these effects may be moderated by relationship quality and dyadic coping style.”

Besides, we have revised our hypotheses to avoid deterministic predictions about the directionality of co-rumination’s effects on synchrony. Instead, our updated hypotheses adopt a more nuanced and open-ended approach, acknowledging that the effects of co-rumination may depend on contextual and relational factors, such as the quality of the relationship and dyadic coping styles.

Timescale Hypotheses

Phasic 1. Relationship partners will display significant levels of movement and cardiovascular synchrony during the emotional sharing tasks, relative to a relevant baseline level of randomized pseudo-interactions.

2. Compared to natural sharing, co-rumination may be associated with enhanced movement and cardiovascular synchrony.

Tonic 3. Emotional sharing will generally make sharers and supporters feel better, leading them to report positive mood changes.

4. Emotional sharing will generally help sharers to reappraise their feelings, leading them to report changes in emotional appraisals.

5. Compared to natural sharing, co-rumination may lead to equal or even more positive mood outcomes, possibly due to increased emotional closeness and perceived support.

6. Compared to natural sharing, co-rumination will be associated with smaller changes in emotional appraisals.

Chronic 7. Higher relationship quality and more constructive dyadic coping strategies will be associated with 1) stronger movement synchrony; 2) stronger cardiovascular synchrony; 3) stronger positive mood changes; 4) more emotional appraisal changes.

8. Higher relationship quality and constructive dyadic coping may buffer or attenuate potential differences between co-rumination and natural sharing in: (a) movement synchrony, (b) cardiovascular synchrony, (c) mood responses, and (d) emotional appraisal changes.

We thank the reviewer again for pointing us to these timely and important contributions, which have meaningfully informed both our theoretical framing and our empirical approach.

DiGiovanni, A. M., Peters, B. J., Tudder, A., Gresham, A. M., & Bolger, N. (2024). Physiological synchrony in supportive discussions: An examination of co‐rumination, relationship type, and heterogeneity. Psychophysiology, 61(7), e14554.

Lin, L., Feldman, M. J., Tudder, A., Gresham, A. M., Peters, B. J., & Dodell-Feder, D. (2023). Friends in sync? Examining the relationship between the degree of nonverbal synchrony, friendship satisfaction and support. Journal of Nonverbal Behavior, 47(3), 361-384.

Vacharkulksemsuk, T., & Fredrickson, B. L. (2012). Strangers in sync: Achieving embodied rapport through shared movements. Journal of experimental social psychology, 48(1), 399-402.

- It was not clear until a while into the introduction that the chronic timescale would not be assessed as a follow up to the conversation, but instead as variables that affect that in-person conversation. This should be made clearer.

Our response: We thank the reviewer for this helpful comment. We agree that the distinction between how the chronic timescale is conceptualized and measured in our study could be more explicitly stated early on. To address this, we have revised the Abstract and the Introduction.

Specifically, we now clarify that:

“At the chronic timescale, the study will primarily assess individual differences in relationship quality and dyadic coping style prior to the task, which are expected to shape phasic and tonic patterns during emotional sharing.” (Revised Abstract)

Additionally, in the Introduction, we now more directly note:

“Importantly, in this study, the chronic timescale is not assessed longitudinally after the task, but is operationalized through individual and relational characteristics (e.g., dyadic coping and relationship quality) measured before the emotional sharing interaction, which are hypothesized to moderate the effects observed at the phasic and tonic levels.”

We hope this clarifies the design of the study and improves the conceptual coherence for the reader. We thank the reviewer again for pointing out this opportunity to clarify.

- Why do you hypothesize that there will be better mood after co-rumination? Tudder and colleagues did not find this in their paper using the same exact method. The literature on this generally points to worse mood, however digiovanni et al 2021 have shown this to be heterogeneous and the pre-print I mention above that uses state space grids showed no associations with negative mood. So, although there is some support for what you are saying, most co-rum work shows that co-rum is related to worse mood. This needs to be fleshed out more why you would expect this effect.

Our response: We thank the reviewer for highlighting the mixed and sometimes conflicting evidence regarding mood outcomes following co-rumination. We acknowledge that.

In response, we have revised the Introduction to more clearly acknowledge this complexity. We now discuss how co-rumination may simultaneously foster emotional closeness and perceived support, which could result in short-term mood benefits, even if long-term outcomes are less favorable. We also note the importance of distinguishing between subjective feelings of mood improvement and objective regulatory or cognitive benefits. Accordingly, we have modified our hypotheses to reflect a more exploratory and open-ended stance. Instead of positing a definitive mood enhancement following co-rumination, we now hypothesize that co-rumination may lead to equal or potentially better mood outcomes than natural sharing due to increased emotional closeness—while recognizing this effect may be moderated by relationship quality and dyadic coping strategies. These changes are now reflected both in the main text and in the updated hypothesis table.

We are grateful for the reviewer’s insightful critique, which allowed us to present a more nuanced theoretical rationale and hypothesis structure.

- Why not look at any type of relationship outcomes in the tonic phase, like closeness or responsiveness? After reading, it looks like you have those variables, but not hypotheses about them. Why? Or maybe you plan to look at this, but didn't include that in the manuscript.

Our response: We appreciate the reviewer’s thoughtful observation regarding the relationship outcome variables from our list of confirmatory hypotheses.

Indeed, we acknowledge that these variables are empirically relevant to both tonic and chronic interpersonal processes, and we have included them in our measurement framework (as detailed in the Supplementary Materials). However, we decided not to formulate formal hypotheses regarding these variables due to the more exploratory nature of our investigation into how they relate to the main outcomes (e.g., synchrony, mood, and appraisal change). Given the novelty and complexity of the multiscale framework employed in our study, we opted to prioritize theoretically grounded and statistically parsimonious hypotheses in the main manuscript.

That said, we recognize the importance of these relational outcomes and plan to include them in our exploratory analyses. We have now clarified this rationale in the manuscript and highlighted these variables more explicitly as part of our exploratory aims in the supplementary section. Once again, we thank the reviewer for encouraging us to reflect on the broader scope of our outcome measures and to ensure that our manuscript clearly communicates the full analytical plan.

- I worry about having participants engage in 4 conversations. Two conversations is already taxing enough, and I dont think it is a good idea to have people do both co-rum and natural sharing. I would have suggested to keep people in one condition. However, it seems that data collection is finished now. I might instead analyze data only for the first condition for all participants.

Our response: We thank the reviewer for this thoughtful and important observation. We acknowledge that engaging in four emotionally intense conversations might be taxing for participants.

At the time of the study design, we opted for a within-dyad design in which all participants experience both the co-rumination and natural sharing conditions. This design was chosen for two main reasons: (1) to control for individual differences in emotional sharing styles and interpersonal dynamics across dyads, thereby increasing the statistical power and reducing the influence of between-dyad variability; (2) to allow for a direct within-subject comparison of co-rumination and natural sharing effects under ecologically valid conditions, while maintaining a counterbalanced and randomized orde

---

## [Editor Report · Decision Letter 1]

10 Apr 2025

From dyadic coping to emotional sharing and multimodal interpersonal synchrony: Protocol for a laboratory experiment

PONE-D-24-48074R1

Dear Dr. Zeng,

We’re pleased to inform you that your manuscript has been judged scientifically suitable for publication and will be formally accepted for publication once it meets all outstanding technical requirements.

Kind regards,

Rachel Shu Tyng Low

Academic Editor

PLOS ONE
---

## [Editor Report · Acceptance letter]

PONE-D-24-48074R1

PLOS ONE

Dear Dr. Zeng,

I'm pleased to inform you that your manuscript has been deemed suitable for publication in PLOS ONE. Congratulations! Your manuscript is now being handed over to our production team.

Kind regards,

on behalf of

Dr. Rachel Shu Tyng Low

Academic Editor

PLOS ONE